# Differences in stiffness across the patellar tendon: An observational study using tendotonometry

Lotte van Dam[1,2*], Rieneke Terink[2], Inge van den Akker-Scheek[3], Johannes Zwerver[1,2]

1 Center for Human Movement Sciences, University of Groningen, University Medical Center Groningen, Groningen, The Netherlands, 2 Sports Valley, Department of Sports Medicine, Gelderse Vallei Hospital, Ede, The Netherlands, 3 Department of Orthopedics, University of Groningen, University Medical Center Groningen, Groningen, The Netherlands

* c.h.j.van.dam@umcg.nl

## Abstract

Patellar tendinopathy is an injury with pain mostly located at the proximal region of the tendon and possibly related to changes in patellar tendon (PT) stiffness. However, little is known about the stiffness at different locations within the PT and the reliability of measuring stiffness at these locations. Therefore, the aim of this study was to investigate the difference in PT stiffness and reliability of its measurement at different PT locations in non-injured male (n = 24) and female (n = 24) recreational athletes. Tendotonometry was performed in triplet at nine locations on the PT of the dominant leg (proximal, horizontal-midline (HM) and distal, at all three levels: medial, vertical-midline (VM) and lateral) in 90 degrees knee flexion in the resting position. Both in males and females, PT stiffness was higher proximal and distal compared to HM and lower medial and lateral compared to VM. PT stiffness ranged from 702.3 ± 124.6N/m to 882.6 ± 117.7N/m for females and 822.6 ± 124.7N/m to 990.3 ± 141.0N/m for males. With respect to reliability, the intraclass correlation coefficient was above 0.9, and the coefficient of variation was below 3% at all nine measurement locations, indicating excellent reliability. In conclusion, PT stiffness is higher at the proximal and distal part of the tendon compared to HM, and lower at the medial and lateral part of the tendon compared to VM. Tendotonometry can reliably be used to measure PT stiffness at different locations on the tendon.

## Introduction

Tendons are structures that connect muscle to bone, generating joint movement [1,2]. One such example is the patellar tendon (PT), which can be considered a bone-to-bone attachment, attached to the patella and tibia [3,4].

The PT is exposed to load regularly during activities of daily life and work but also sports and other physical activities [5,6]. It adapts well and improves its load capacity

**Data availability statement:** All relevant data are within the paper and its Supporting Information files.

**Funding:** The author(s) received no specific funding for this work.

**Competing interests:** All authors affirm that they have no affiliation (including research funding) or involvement with any commercial organization that has a direct financial interest in any matter included in this manuscript.

when there is adequate load combined with a sufficient recovery period [7,8]. However, under excessive loading and insufficient recovery, maladaptation makes the PT susceptible to injury [5,9], such as patellar tendinopathy, also known as the jumper's knee [10–12]. The prevalence of patellar tendinopathy among elite basketball and volleyball players ranges up to 31.9% and 45% respectively [10,11]. Patellar tendinopathy can result in prolonged or recurring symptoms, which negatively impact sports performance, activities of daily life, and work [13,14]. Three distinct variants of patellar tendinopathy are known: insertional quadriceps tendinopathy, distal patellar tendinopathy, and proximal patellar tendinopathy [15–20]. Insertional quadriceps tendinopathy is reported in 25% of the cases and is located above the patella. This part of the tendon has a different anatomy and function compared to the region below the patella, and therefore also has a distinct etiology and treatment [21]. Distal patellar tendinopathy is responsible for 10% of patellar tendinopathy cases and is located at the most distal end of the tendon, attached to the top of the tibial tubercle. The most common variant of patellar tendinopathy is proximal patellar tendinopathy (also known as patellar apexitis, due to its origin at the patellar apex), which is reported in 65% of cases [20]. These three variants of patellar tendinopathy all arise at the insertions; midtendon tendinopathy rarely occurs. The cause of patellar tendinopathy is still unclear and factors such as tendon health, sex, age, weight, engagement in exercise and type of sport seem to have an influence on its development [22–28].

Studies observing the mechanical properties of pathological tendons have observed alterations in stiffness compared to healthy tendons [29–32], suggesting there is a relationship between tendon stiffness and (the development of) symptoms. Measuring tendon stiffness on a regular basis in athletes at the location of the injury could inform coaches and medical staff about the health status of an athlete's tendons and might therefore be helpful in the prevention and rehabilitation of tendinopathy. Several stiffness measurement techniques exist, such as shearwave elastography, ultrasound-based calculations, and MRE measurements. However, many of these measurements require technical expertise, can be time consuming, are not portable, and can be expensive. Tendotonometry – the measurement of compressive transverse tendon properties with a handheld digital palpation device – might be a more feasible method for measuring tendon stiffness on a regular basis in athletes [33]. Indeed, existing research has shown promising results in the measurement of Achilles and PT stiffness [34–40].

In most studies performed on PT stiffness, measurements of stiffness were done at the midpoint of the PT [35,40–44], or the exact measurement location was not mentioned [45]. To the best of our knowledge, only two other studies investigated variation in stiffness across the PT at other locations [46,47]. Although both studies also investigated PT stiffness using tendotonometry, only one investigated the reliability of this method, and did so at only three locations (proximal-middle-distal) [46]. Additionally, athletes and females were excluded [46]. While the other study included athletes (gender unknown) and measured at six different locations below each other from proximal to distal, no reliability analyses were performed [47]. Altogether, stiffness and reliability data on a wider area across the PT (also in medial and lateral

directions) is still lacking for males and females separately. Since the prevalence of patellar tendinopathy, especially at the proximal region of the tendon, is higher in the athletic population, it is important to know how tendon stiffness varies in healthy athletes and if this can be reliably measured across the tendon. If so, tendon stiffness can be monitored at the exact location of the injury to identify changes compared to the healthy situation, and to understand how injury develops. In this context, this study aimed to investigate if PT stiffness and the reliability of its measurement differ across nine different locations of the PT in recreational athletes. We hypothesize that i) PT stiffness can be measured reliably at different locations across the tendon, and ii) that PT stiffness varies across the tendon, with an increasing stiffness closer to the insertions.

## Materials and methods

### Study setting

An observational single-center study was conducted at Sports Center de Bongerd, Wageningen University & Research Campus. All measurements were performed in the meeting room of the sports center between 14 May 2024 and 4 July 2024 in the late afternoon and evening. Athletes did not perform any sports activities at least within one hour of measurements. Travel time and mode of transport to the sports center were enquired to ensure that participants did not engage in any type of active travel (for example walking, cycling or running) before the measurements. Participants then completed a short questionnaire after which PT stiffness measurements were performed.

### Ethics

The study was conducted according to the principles of the Declaration of Helsinki (64th WMA Assembly, October 2024). Only observational coded data were used. This study received a non-WMO declaration from the Medical Ethics Committee Oost-NL on the 13th of April 2023 (Approval No. 2023(16339)), as the Dutch law on Research Involving Human Subjects Act (WMO) did not apply.

### Participants

Participants were recruited via the sports associations of Wageningen University and Research. We contacted family and friends and used social media platforms of the sport associations for recruitment. All participants who were interested received an information letter about the study protocol. Participants could combine the measurements before their planned sports session on their preferred day. Healthy male and female athletes aged between 16 and 40 years with a BMI between 18.5 and 25 kg/m² and who performed exercise more than once a week were eligible. Participants were excluded if they experienced knee problems, or if they took medication that could affect musculoskeletal function. No other inclusion- or exclusion criteria were used. All participants provided written informed consent before participation.

### Study procedure

**Questionnaires and anthropometric measurements.** Participants completed a short questionnaire to obtain information regarding age, type of sport, amount of training hours, history of knee injuries, and lower extremity dominance. In order to rule out the likelihood of PT problems, participants filled out the Victorian institute of sport assessment – patella (VISA-P) questionnaire [48,49]. This is a short and simple questionnaire specifically designed for athletes with patellar tendinopathy which measures the severity of the injury by assessing pain, function, and ability to participate in sport activities. A VISA-P score of 100 indicates no pain, maximum function and maximum ability to play sports, whereas a VISA-P score of 70 indicates symptoms of patellar tendinopathy.

In addition to the questionnaires and PT stiffness measurements, the height and weight of the participants were measured on site by trained staff. Measurements took place clothed but without shoes on.

**PT stiffness measurements.** All stiffness measurements were performed with the MyotonPRO (device code 1308600502, SN000041, Tallinn, Estonia). The MyotonPRO is a non-invasive handheld device for measuring muscle, tendon, and other soft tissue properties. The device applies a brief pulse to the skin overlying the tendon; thereafter, several oscillation parameters are used to calculate the mechanical properties of the tissues, including compressive tendon stiffness [29,50]. PT stiffness was measured at a knee angle of 90° defined using a goniometer with the participant in supine position and the feet placed on the examination table (Fig 1).

Stiffness was measured at nine different locations on the tendon, as shown in Fig 2. The upper row of measurement locations was defined as 10% of tendon length (proximal), the middle row at 50% (horizontal midline, HM) and the lower row at 90% (distal). This is also the case for the width of the tendon (10% of tendon width being medial, 50% of tendon width being vertical midline (VM) and 90% of tendon width being lateral), resulting in nine different locations (Fig 2).

Three consecutive measurements were performed at each location. One experienced operator defined and marked all locations on all test days, and another performed all measurements on the PT of the dominant leg. The probe of the MyotonPRO was held perpendicular to the skin overlying the PT (±5°, monitored by the device itself), with a deformation area of 7.1mm². Next, the probe was pushed against the skin to reach the correct depth. This was signaled by a red light turning green on the MyotonPRO, indicating a pre-compression strength of 0.18N. This was followed automatically by five short impulses of 0.4N with a tap interval of 0.8 seconds to induce mechanical damped oscillations in the underlying tissues. The MyotonPRO provides mean values on the dynamic stiffness (S, N/m) of the five impulses delivered. This stiffness value is calculated by the device with the maximum acceleration of the oscillation and the deformation of the tissue detected by the transducer. It is important to note that the MyotonPRO measures transverse mechanical properties of the tendon, reflecting its compressive stiffness perpendicular to the collagen fiber direction, rather than the tensile stiffness along the tendon's longitudinal axis.

## Statistical analysis

A priori type of power analysis and sample size calculation were performed with G*Power (version 3.1.9.7). The sample size of 22 per group (males and females separately) was calculated in order to achieve a power (1-β err prob) of 0.95 with an effect size of 0.25 and $\alpha = 0.05$, with a predefined minimal detectable change (MDC) of 40.3N/m [51].

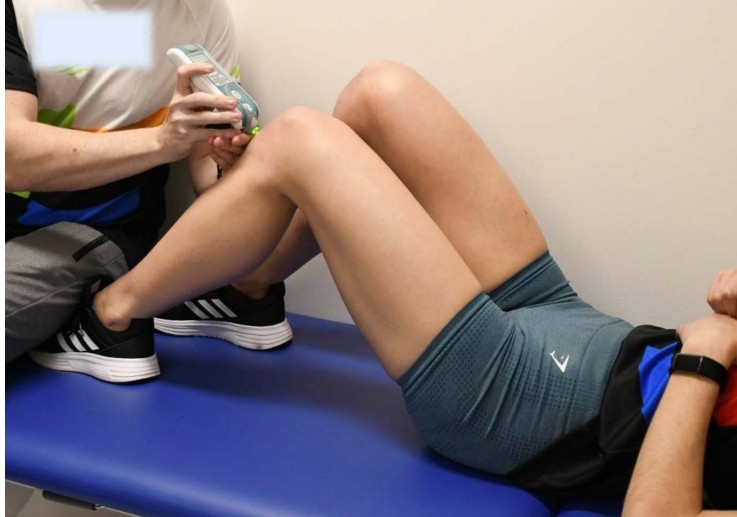

**Fig 1. The position of the participants while performing tendotonometry.**

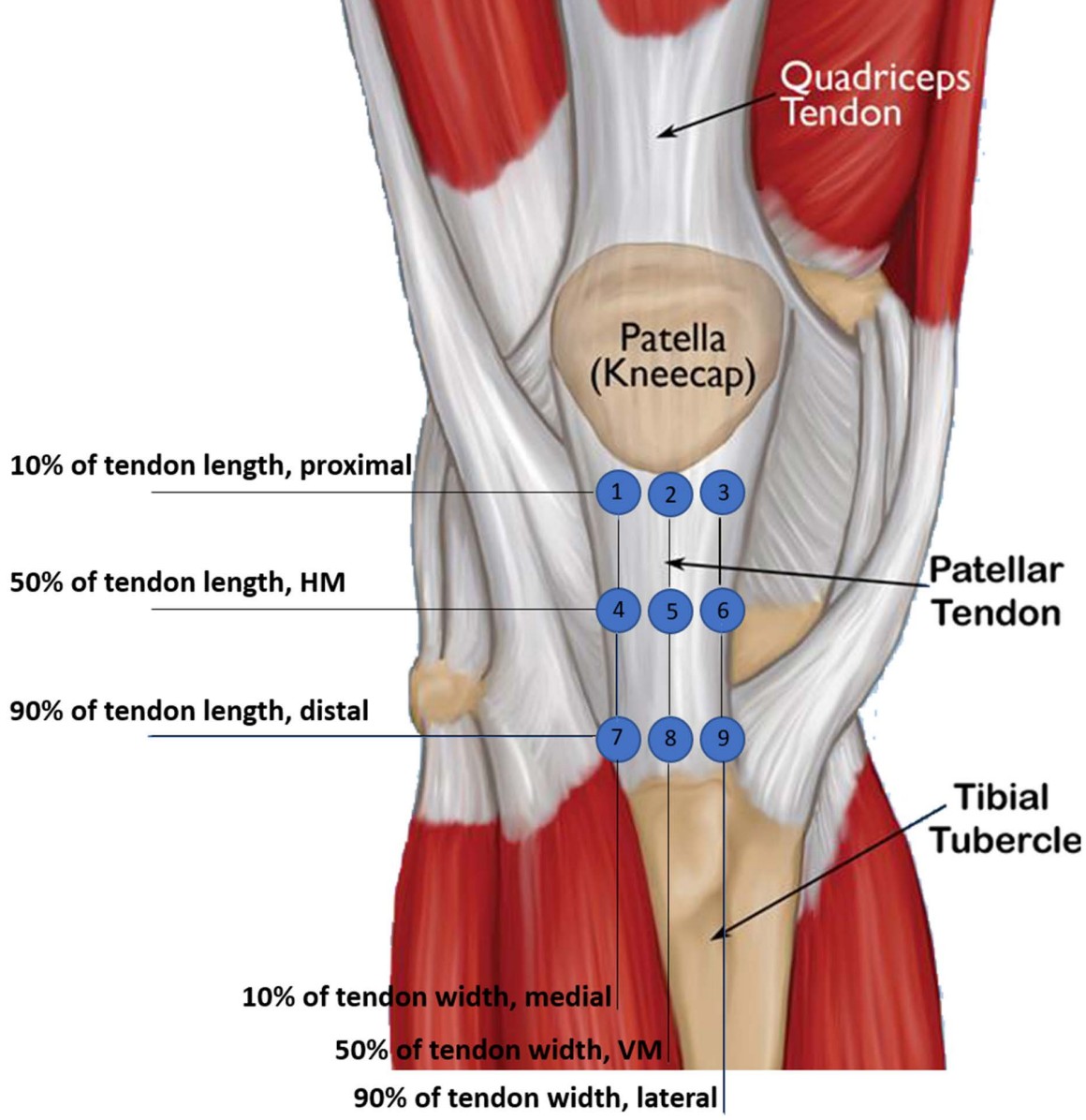

**Fig 2. The nine different measurement locations on the tendon of the left knee as example.** 1) proximal-medial, 2) proximal-HM, 3) proximal-lateral, 4)VM-medial, 5) VM-HM, 6) VM-lateral, 7) distal-medial, 8) distal-HM, 9) distal-lateral.

Results were analyzed using IBM SPSS Statistics 24 (Armonk, New York, USA). Demographic characteristics of the participants were assessed through descriptive statistics. Normality was checked with the Shapiro-Wilk test and visually assessed using histograms. As the data were normally distributed, they are presented as mean ± SD or graphically with 95% CI, with the use of Microsoft Excel 2016.

To investigate the difference in stiffness within the nine different locations, the average PT stiffness of the threefold measurements per location was used in the one-way ANOVA. Tukey HSD test was used as post-hoc test to test for specific group differences. These differences were investigated only for the dominant PT and separately for males and

females, since PT stiffness is known to significantly differ between sexes [51]. Statistical significance of all tests was set at p < 0.05.

To investigate the reliability of the nine different locations, intraclass correlation coefficients (ICCs) were calculated between the three measurements per participant that were taken at each location on the tendon (2,1, two-way random model, single measures). Here, males and females were combined since the literature showed no difference in sex with respect to measurement reliability [51]. Reliability was considered excellent when ICC values exceeded 0.75, good-to-fair for ICCs between 0.40 and 0.75, and poor for ICCs below 0.40 [26]. Coefficient of variation (CV) was calculated for all measurements by dividing the SD by the mean of the three measurements taken per measurement location, per participant. A maximum CV of 3% was defined as good [27], and therefore we calculated the amount and percentage of measurements above and below this predefined cutoff value.

## Results

### Basic characteristics

Forty-eight athletes participated in this study (24 males, 24 females, Table 1). The participants of the current study performed various sports, including fitness (n = 15) and team sports (n = 12) (S1 Table).

### Differences in stiffness across the tendon

In both males and females, PT stiffness was higher at both the proximal and distal sites compared to the HM. PT stiffness at the VM was higher than the medial and lateral regions (Tables 2, S2 and S3). The difference in stiffness between the HM and proximal was larger than between HM and distal. This pattern was not observed when comparing VM with medial and lateral regions.

**Table 1. Summary of basic characteristics.**

| Mean ± SD | Males (n = 24) | Females (n = 24) | Total (n = 48) |
|---|---|---|---|
| Age (years) | 23.0 ± 3.9 | 24.2 ± 4.9 | 23.6 ± 4.4 |
| Height (cm) | 183 ± 7 | 171 ± 7 | 178 ± 10 |
| Weight (kg) | 74.7 ± 10.5 | 64.7 ± 7.3 | 69.7 ± 10.3 |
| Body mass index (kg/m2) | 22.1 ± 2.8 | 22.1 ± 2.5 | 22.1 ± 2.6 |
| Dominant leg, Left/Right | 3 L to 21 R | 4 L to 20 R | 7 L to 41 R |
| Training hours per week | 5.0 ± 2.7 | 3.6 ± 2.1 | 4.3 ± 2.5 |
| VISA-P score | 94 ± 7 | 92 ± 13 | 93 ± 10 |

**Table 2. Average PT stiffness (N/m) of the dominant leg.**

| Knee | Location | Medial | VM | Lateral |
|---|---|---|---|---|
| Females | Proximal | 739.3 ± 149.2* | 882.6 ± 117.7 | 815.3 ± 131.3 |
| | HM | 702.3 ± 124.6 | 804.3 ± 130.0 | 711.2 ± 124.6 |
| | Distal | 783.1 ± 112.4 | 825.9 ± 124.4 | 771.2 ± 104.2 |
| Males | Proximal | 864.7 ± 109.3* | 990.3 ± 141.0 | 898.1 ± 108.8 |
| | HM | 832.0 ± 111.7 | 923.3 ± 120.3 | 822.6 ± 124.7 |
| | Distal | 896.1 ± 84.6 | 924.2 ± 105.7 | 854.0 ± 90.6 |

HM: horizontal midline, VM: vertical midline.

*significantly different from VM, analyzed using a One-Way ANOVA.

## Reliability variation over the tendon (intra-operator measurements)

All ICC values, including the confidence intervals, were above 0.962, showing excellent reliability (Table 3, see ICC results separated for males and females S4 and S5 Tables). With respect to the CV, all averages were well below 3%, with 79% or more separate cases below 3% (Table 4).

## Discussion

The current study aimed to investigate differences in PT stiffness and reliability of tendotonometry at various locations on the PT in recreational athletes. In line with our hypothesis, this study showed that PT stiffness was higher at proximal and distal ends compared to HM and lower at the medial and lateral sides compared to VM. We also observed that PT stiffness was higher at the proximal site compared to distal. Our findings also support tendotonometry as a reliable method to measure PT stiffness at different locations on the tendon.

### Differences in stiffness across the tendon

Tendotonometry showed differences in PT stiffness. Although not statistically significant, the differences in PT stiffness between the various locations were above the minimal detectable change of 39.5N/m [51]. Two other papers have previously investigated PT stiffness at multiple locations on the tendon [46,47]. In one study, PT stiffness was measured in 30 healthy male non-athletes without tendon injuries. Stiffness was measured using tendotonometry at three different locations: proximal, HM and distal, with proximal and distal at the very ends of the tendon. The lowest tendon stiffness was found at HM (620.2±105.7), with a higher stiffness proximal (821.1±117.1) compared to distal (714.5±123.7). This variation in stiffness across the tendon is in line with the findings of the current study. Although it was not mentioned which knee was measured, stiffness values were quite lower compared to the participating males in the current study, with a larger variation in stiffness between the measurement points. This could be due to the different study population (athletes versus non-athletes [52–54]) and the different measurement locations (at 10% or at the proximal and distal ends of the tendon). Although the direction of change remains unclear, it is known that athletes have an altered PT stiffness compared to non-athletes, and that one single training session can already change tendon stiffness [33]. Loading can lead to adaptations in tendon stiffness by stimulating collagen synthesis and crosslinking between collagen molecules [55–57], making the tendon stronger and, increasing the load capacity.

The second paper that measured PT stiffness in fifteen healthy elite track cyclists across the tendon using tendotonometry measured at six different locations on the tendon, in a vertical line, 5mm apart, from proximal to distal [47]. Although no statistical tests were performed between the measurements, a repeated measures ANOVA showed a significant effect

**Table 3. ICC values (95%CI) of the threefold measured stiffness.**

| Location | Medial | VM | Lateral |
|---|---|---|---|
| Proximal | 0.962 (0.940-0.977) | 0.977 (0.963-0.986) | 0.967 (0.947-0.980) |
| HM | 0.973 (0.955-0.984) | 0.990 (0.982-0.994) | 0.979 (0.966-0.987) |
| Distal | 0.976 (0.960-0.986) | 0.985 (0.976-0.991) | 0.962 (0.939-0.977) |

**Table 4. Average±SD CV (%) and nr. of cases above 3%.**

| | CV (%) | Nr. above 3%, amount (%) | CV (%) | Nr. above 3%, amount (%) | CV(%) | Nr. above 3%, amount (%) |
|---|---|---|---|---|---|---|
| Location | Medial | | VM | | Lateral | |
| Proximal | 2.2±2.0 | 10 (21%) | 1.3±1.2 | 4 (8%) | 1.7±1.6 | 9 (19%) |
| HM | 2.0±1.3 | 10 (21%) | 1.2±0.7 | 2 (4%) | 1.8±1.4 | 7 (15%) |
| Distal | 1.4±1.1 | 6 (13%) | 1.3±0.7 | 1 (2%) | 1.7±1.4 | 7 (15%) |

of location. However, in contrast to our findings, the highest PT stiffness was not found proximal and distal, but rather 5 mm below the proximal measurement. Although nothing can be said about significance, this difference could be due to a smaller sample size or the different study population.

Differences in stiffness across the tendon might be explained by its anatomical structure. The PT is attached to bone both at the inferior pole of the patella and the anterior tibial tubercle. This tendon-to-bone insertion facilitates a gradual transition of force based on different histological and mechanical properties of tendon and bone tissue to avoid local peaks in tension [58,59]. Therefore, the tendon tissue gradually transitions into bone at both sides of the tendon. The insertions of the patella tendon are also known as fibrocartilaginous insertions, where the gradual transition from tendon to bone is divided into four zones of tissue: dense fibrous connective tissue (e.g., tendon), uncalcified fibrocartilage, calcified fibro-cartilage, and bone [58]. This gradual transition could explain the higher stiffness proximal and distal with respect to HM.

It is a general mechanical principle that stress is located at transitions between structures containing different mechanical properties [60]. Despite this gradual transition from tendon to bone like structure, the insertions are still more vulnerable to stress compared to the rest of the tendon. This could also explain why the proximal insertion is the most commonly affected anatomical site in patients with patellar tendinopathy, and why all variants of patellar tendinopathy are located at the insertions of the tendon [15–19]. This proximal region is the location of the tendon with the highest stiffness, and it has been suggested that higher PT stiffness is related to patellar tendinopathy [31,61–63]. Hence, the proximal measurement location of the PT seems to be the most relevant location to monitor stiffness.

A hypothesis that could potentially explain the lower stiffness medial and lateral compared to the VM is the leverage hypothesis. When medial and lateral are compared with VM, the outer sides of the tendon are reached together with soft connective tissue. This is explained mechanically using the leverage effect. At the outer sides of an object (the tendon in this case), a higher moment is reached with the same force (the taps of the tendotonometer) due to the longer arm (distance from midpoint tendon to the outer side). Due to this higher moment, more movement is generated, and thus the tendon is less stiff at the outer sides. Another hypothesis that is that the core of a tendon demonstrates the lowest (if not negligible) tissue renewal [64]. With tissue adaptations occurring at the outer regions of a tendon, the core might show a higher stiffness compared to these newer, less stiff, outer regions of a tendon. However, one point against this hypothesis is that this often holds true for smaller tendons, whereas the PT is a large and thick tendon. A final hypothesis that could explain the lower stiffness medial and lateral compared to VM might be the variation in thickness of the tendon [65]. Ito et al. found variation in PT thickness across the tendon, with the PT being especially less thick at the lateral region compared to VM [65]. With the PT being less thick at the outer regions, resistance to a certain force might also be lower, resulting in lower PT stiffness.

### Variation in reliability over the tendon

With excellent ICCs and a mean CV below 3%, PT stiffness can be measured reliably at all nine measurement locations. Although the ICC values were highest and the CV values lowest at midpoint (VM/HM), all results showed excellent reliability. Only one other article could be found investigating the reliability of different locations on the PT to measure tendon stiffness [46]. In this study, the reliability of PT stiffness tendotonometry was measured at three locations – proximal, HM and distal – and both ICC and CV were investigated. In agreement with our findings, this study also found excellent reliability at all three measurement locations [46].

As stated above, the proximal location on the PT might be an interesting location to measure stiffness since this is a location that is frequently affected by tendinopathy. Since reliability appeared to be excellent, our results suggest that researchers, coaches, health care professionals, and other potential tendotonometry users can safely use this method to measure tendon stiffness across the tendon, particularly at the proximal region. However, one note of attention is that when stiffness is measured at a location close to bone (e.g., the patella), excessive stiffness can lead to errors in tendotonometry measurements, which might increase the time taken to obtain accurate readings.

## Strengths and limitations

A strength of the current study is the inclusion of a broad range of athletes. Both males and females were included, in contrast to other studies which often exclude females [41–43,46]. Additionally, fourteen types of sports were performed by the total group of participants, demonstrating a broad array of PT loading and thus considerable variation in PT stiffness due to sex, age, and type of sports [23,66–68]. This gives us insight into the PT stiffness measurement possibilities amongst healthy recreational athletes involved in different activities. However, more research is necessary to investigate the variation in PT stiffness within the tendon and its reliability in a population with patellar tendinopathy. A second strength of the current study is that stiffness measurements were performed in a standardized manner, since research showed that positioning of the subject might influence stiffness outcomes and reliability [51]. Finally, the operator was blinded to the study results since all results were recorded by a distinct researcher, increasing the methodological quality.

This study also has some limitations. A first limitation is that only little is known about the validity of the MyotonPRO as a measurement method. One article could be found investigating its validity, in which it was compared to shear-wave elastography (SWE) [37]. Validity appeared to be good, however, was only investigated at the achilles tendon instead of the PT. Therefore, more research is needed before the validity of the measurement method can be guaranteed. A second limitation of the current study is that muscle activity was not monitored during the PT stiffness measurements. Although participants were instructed to lie down with fully relaxed muscles, PT stiffness could have varied due to muscle activity [69]. In future research projects, we propose that muscle activity should be monitored using EMG measurements to guarantee full muscle relaxation [70,71]. Since this increases the complexity of the measurement execution and diminishes the ease of the PT stiffness measurement itself, this is only recommended in a research setting. A third limitation is that athletes using oral contraceptives were not excluded from participation, since this would have led to the exclusion of a large part of the female athlete population. However, it is known that oral contraceptives can influence tendon properties and collagen synthesis, and therefore this may have influenced our findings [72]. Finally, PT stiffness measurements using the MyotonPRO quantify tendon stiffness perpendicular to the PT, while the actual tendon and its stiffness run longitudinal. We are not certain how compressive, transverse measures of stiffness relate to the ability of tendons to withstand tensile deformation. Resent evidence might suggest that compressive stiffness measured using the MyotonPRO does not reflect the tendon's tensile stiffness under load, as Ishigaki et al. (2025) found no significant relationship between the two measures in the achilles tendon [73]. However, as this study compared measurements in rest (compressive stiffness) with measurements during exercise (tensile stiffness) and these measurements took place in a different tendon, extrapolation to the PT should be done with caution. Nevertheless, this study is proof that care is necessary when drawing conclusions about longitudinal versus perpendicular tendon stiffness.

## Recommendations

For practical use:

- PT stiffness is higher at the proximal region compared to the horizontal midline and distal in recreational athletes. This is also the region where patellar tendinopathy occurs most frequently. We recommend using measurements taken at the proximal region.

- Tendotonometry might be a promising tool to measure tendon stiffness at different locations, including the proximal location on the tendon, which is most often affected.

Future studies might investigate:

- The validity of tendotonometry at different locations on the PT.

- The variation in PT stiffness and its reliability in participants with patellar tendinopathy.

- The variation in PT stiffness variation over time, for example during a training season. We now know PT stiffness can be measured reliably at different locations on the tendon, though less is known about variation in PT stiffness over time. Understanding this would facilitate injury detection at an earlier stage.

## Conclusion

In recreational athletes, PT stiffness was higher proximal and distal compared to the horizontal midline and lower from the vertical midline towards the medial and lateral sides. Tendotonometry might be a promising tool to measure PT stiffness at different locations on the tendon.

## Supporting information

**S1 Table. Sports performed by participants.**
(DOCX)

**S2 Table. Tukey HSD posthoc comparisons of the nine different measurement locations within the female patellar tendon.** 1) proximal-medial, 2) proximal-horizontal midline, 3) proximal-lateral, 4)vertical midline-medial, 5) vertical midline-horizontal midline, 6) vertical midline-lateral, 7) distal-medial, 8) distal- horizontal midline, 9) distal-lateral. For the right knee, medial and lateral are reversed.
(DOCX)

**S3 Table. Tukey HSD posthoc comparisons of the nine different measurement locations within the male patellar tendon.** 1) proximal-medial, 2) proximal-horizontal midline, 3) proximal-lateral, 4)vertical midline-medial, 5) vertical midline-horizontal midline, 6) vertical midline-lateral, 7) distal-medial, 8) distal- horizontal midline, 9) distal-lateral. For the right knee, medial and lateral are reversed.
(DOCX)

**S4 Table. ICC values (95%CI) of the threefold measured stiffness specified for females.**
(DOCX)

**S5 Table. ICC values (95%CI) of the threefold measured stiffness specified for males.**
(DOCX)

**S1 Dataset. The complete dataset of the study with which all data analyses have been performed.**
(XLSX)

## Acknowledgments

We would like to thank the staff of the Bongerd sports center for providing us a room to perform all the measurements. We thank Dan Kirk for improving the language of the article.

## Author contributions

**Conceptualization:** Lotte van Dam, Rieneke Terink, Johannes Zwerver.

**Data curation:** Lotte van Dam.

**Formal analysis:** Lotte van Dam.

**Investigation:** Lotte van Dam, Rieneke Terink.

**Methodology:** Lotte van Dam, Rieneke Terink, Johannes Zwerver.

**Resources:** Johannes Zwerver.

**Supervision:** Rieneke Terink, Inge van den Akker-Scheek, Johannes Zwerver.

**Visualization:** Lotte van Dam.

**Writing – original draft:** Lotte van Dam.

**Writing – review & editing:** Rieneke Terink, Inge van den Akker-Scheek, Johannes Zwerver.

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
