## [Decision Letter · Decision Letter 0]

21 May 2025

PONE-D-25-19089Differences in stiffness across the patellar tendon. An observational study using tendotonometry.PLOS ONE

Dear Dr. van Dam,

Thank you for submitting your manuscript to PLOS ONE. After careful consideration, we feel that it has merit but does not fully meet PLOS ONE’s publication criteria as it currently stands. Therefore, we invite you to submit a revised version of the manuscript that addresses the points raised during the review process.

We look forward to receiving your revised manuscript.

Kind regards,

Charlie M. Waugh

Academic Editor

PLOS ONE

Journal Requirements:

2. We are unable to open your Supporting Information file [Dataset.sav]. Please kindly revise as necessary and re-upload.

Reviewers' comments:

Reviewer's Responses to Questions

**Comments to the Author**

1. Is the manuscript technically sound, and do the data support the conclusions?

Reviewer #1: Yes

Reviewer #2: Yes

2. Has the statistical analysis been performed appropriately and rigorously? 

Reviewer #1: Yes

Reviewer #2: Yes

3. Have the authors made all data underlying the findings in their manuscript fully available?

Reviewer #1: Yes

Reviewer #2: No

4. Is the manuscript presented in an intelligible fashion and written in standard English?

Reviewer #1: No

Reviewer #2: Yes

5. Review Comments to the Author

Reviewer #1: 1. Overall, this manuscript could use a significant amount of work to increase readability to bring it to the scientific standard. Please copy edit for grammar, acronym usage, punctuation, formatting (tables, spacing, references), and writing style (overly verbose at times).

2. I can see the usefulness of devices like the MyotonPRO as a method of quickly assessing a mechanical value of the tendon in a reliable manner. However, when we think of stiffness of a tendon, we typically think of it with regards to the tensile properties along the long axis in line with the collagen fibres/fascicles. I acknowledge that the authors do note that this is a form of compressive stiffness and make mention of the questions surrounding validity of the device in the limitations. Tendons are anisotropic tissues; and currently, we aren’t certain how compressive/transverse measures of stiffness relates to how tendons withstand tensile deformation. The authors should comment on this and add more of a distinction between the two mechanical measures in the paper - perhaps in the methods section and broaden the limitations statement as well.

3. The use of the term "tendotonomtry" is a neologism that is not found in the broader literature beyond the author’s previous publications. I recommend using more widely accepted terminology unless a strong justification is provided. Introducing new terms should be reserved for clearly novel concepts, and their definitions and distinctions from existing methods should be clearly articulated.

4. The authors need to expand on the gap in knowledge in the introduction, why is it important to take measurements in different locations on the PT? Why is it important to distinguish between recreational athletes and non-athletes? Also Zhu 2022 did not include females in their analysis, you could highlight that this is also part of the gap in knowledge.

5. The hypothesis needs to be clearly stated.

6. Line 80-83. “Measurements took place before any sports activities were performed by the athletes.” For that specific day? If the participants exercised, how long after exercise would the researchers wait to take measurements, were participants instructed not to exercise for x-hours/days prior to testing?

7. Line 93. What medications were a part of your exclusion criteria? Does this include medications like oral contraceptives, which have been shown to alter tendon mechanics?

8. How did you determine your sample size? Please provide necessary information.

9. Line 92. Given units for BMI (kg/m2). Was the stated BMI range an exclusion criteria? If yes, 25 kg/m2 is a very low cut-off value for BMI even in a healthy population. Please explain.

10. Lines 139-142 should be before “statistical significance was set at p<0.05”, lines 137-138.

11. Lines 145-146, “Males and females were combined together, since literature showed no difference in gender with respect to reliability of the measurement [50].” The authors could provide ICC analysis for both sexes, since the previous publication did not assess different locations on the PT and you’ve already provided some separate analysis of sex. When assessing biological differences, use sex rather than gender.

12. Table 2. Provide units, identify acronyms, tests used. Please report on the statistical test results. Publications have obligations to provide as much detail as possible so that your results can (potentially) be used for future reviews and field dissemination.

13. Lines 160-163. Same as comment #1, need some edits for readability especially when acronyms are being used. Example: both proximal and distal stiffness were greater than HM, and VM stiffness was greater than medial and lateral.

14. Table S2 could and probably should be presented as Table 3 within the results rather than supporting information. Also recheck ICC values, line 168 states all values were above 0.963, in the table it is 0.962. Perhaps a typo?

15. Lines 169 to 170. This is a little misleading, all cases were not <3%; in S3 you state a % of cases >3%. More clarity needed here. I also suggest adding S3 to the main body of the paper as well, or at least combining it with the previous table?

16. Lines 172-176. Once hypothesis is clarified (#5), a statement here would be appropriate.

17. Lines 188-189, broaden argument about why recreational athletes would have altered stiffness compared to non-athletes.

18. Line 150 “Maximum CV allowed for tendotonometry was preset at 3%”. I assume the authors would retake the exam if the measure was >3%? This adds confusion to lines 224-225 “mean CV below 3%”, the mean CV is always going to be below 3% because of the cut off value. There needs to be more clarity regarding CV and the 3% threshold.

19. General comment, there are a few statements throughout the manuscript that are overly definitive with regards to previous literature and your results; please make according adjustments.

Reviewer #2: The manuscript addresses an important gap in the literature regarding the spatial variation of patellar tendon (PT) stiffness and the reliability of tendotonometry across multiple tendon locations in healthy recreational athletes. The study is well-motivated, and the findings have potential implications for both clinical assessment and injury prevention. However, several aspects require clarification, expansion, or revision to strengthen the manuscript.

However, a few major suggestions were find to increase the quality of this manuscript.

1. Introduction

- The introduction briefly defines tendotonometry but does not clearly differentiate it from other stiffness measurement techniques (e.g., shear wave elastography, ultrasound).

- Authors have not analyzed the entire research field. For instance Klich et al. (2023) investigated ultrasound and myotonometry to characterize patellar tendon properties.

https://pubmed.ncbi.nlm.nih.gov/36690028/

- The gap is stated, but the novelty of examining nine locations (vs. three in previous studies) could be emphasized more.

- The introduction discusses three variants of patellar tendinopathy, but the connection to the current study (which only includes healthy athletes) is not fully clear. Please add some findings to athletes and overuse athletes.

- The introduction discusses three variants of patellar tendinopathy, but the connection to the current study (which only includes healthy athletes) is not fully clear. Please add a clear hypothesis.

2. Material and methods

- Inclusion of both male and female recreational athletes is a strength.

- Provide more detail on inclusion/exclusion criteria and recruitment process.

- Clarify how the three measurements per site were handled statistically.

3. Discussion

- Relate findings to tendon structure and function. Could regional stiffness differences relate to localized injury risk?

- Limitations are not detailed in the provided sections. Please re-edit.

6. PLOS authors have the option to publish the peer review history of their article (what does this mean? ). If published, this will include your full peer review and any attached files.

**Do you want your identity to be public for this peer review?** For information about this choice, including consent withdrawal, please see our Privacy Policy .

Reviewer #1: No

Reviewer #2: No

---

## [Author Response · Author response to Decision Letter 1]

1 Jul 2025

Response to editor and referees of manuscript PONE-D-25-19089

Differences in stiffness across the patellar tendon. An observational study using tendotonometry.

Dear editor,

Thank you for offering the opportunity to revise the manuscript entitled “Differences in stiffness across the patellar tendon. An observational study using tendotonometry.” (PONE-D-25-19089). We are thankful to the reviewers for their useful and constructive feedback, which, we believe, has further improved our manuscript.

Below you will find our reply to the comments, which have been incorporated into the revised manuscript. We are submitting the revised manuscript using tracked changes.

Thank you for your time and effort to re-evaluate our revised manuscript.

We look forward to your decision.

Yours sincerely,

Also on behalf of the co-authors,

Lotte van Dam, MSc

Reviewer #1:

1. Overall, this manuscript could use a significant amount of work to increase readability to bring it to the scientific standard. Please copy edit for grammar, acronym usage, punctuation, formatting (tables, spacing, references), and writing style (overly verbose at times).

1. We thank the reviewer for this feedback to improve the manuscript with more scientific and less verbose writing. We critically ran through the manuscript to change the writing where we considered it necessary, and also based on the suggestions of both reviewers. Additionally we asked an English-language professional to copyedit the manuscript. We hope these adaptations are more in line with the reviewer’s expectations.

2. I can see the usefulness of devices like the MyotonPRO as a method of quickly assessing a mechanical value of the tendon in a reliable manner. However, when we think of stiffness of a tendon, we typically think of it with regards to the tensile properties along the long axis in line with the collagen fibres/fascicles. I acknowledge that the authors do note that this is a form of compressive stiffness and make mention of the questions surrounding validity of the device in the limitations. Tendons are anisotropic tissues; and currently, we aren’t certain how compressive/transverse measures of stiffness relates to how tendons withstand tensile deformation. The authors should comment on this and add more of a distinction between the two mechanical measures in the paper - perhaps in the methods section and broaden the limitations statement as well.

2. We thank the reviewer for mentioning this critical difference. Indeed we find it important to mention that perpendicular tendon stiffness has been measured and that care should be taken in translating this to the longitudinal tendon stiffness. The possibility to withstand deformation in the longitudinal direction probably differs from the perpendicular direction. To further clarify this distinction, we have elaborated the term tendotonometry in the introduction: “Tendotonometry - the measurement of compressive transverse tendon properties with a handheld digital palpation device - might be a more feasible method for measuring tendon stiffness on a regular basis in athletes”, see line 64-66. In addition, we added some sentences in the methods section: “It is important to note that the MyotonPRO measures transverse mechanical properties of the tendon, reflecting its compressive stiffness perpendicular to the collagen fiber direction, rather than the tensile stiffness along the tendon's longitudinal axis.”, line 146-149, and elongated the limitation section in the discussion: “Finally, PT stiffness measurements using the MyotonPRO quantify tendon stiffness perpendicular to the PT, while the actual tendon and its stiffness run longitudinal. We are not certain how compressive, transverse measures of stiffness relate to the ability of tendons to withstand tensile deformation. Thus, caution is necessary when drawing conclusions about longitudinal versus perpendicular tendon stiffness.”, line 303-307.

3. The use of the term "tendotonometry" is a neologism that is not found in the broader literature beyond the author’s previous publications. I recommend using more widely accepted terminology unless a strong justification is provided. Introducing new terms should be reserved for clearly novel concepts, and their definitions and distinctions from existing methods should be clearly articulated.

3. We understand the opinion of the reviewer. We have introduced the term tendotonometry in a different publication since we did not agree with the existing terminology. As the Myoton is developed first and foremost for muscles (as the name also clearly shows), we don’t think the term myotonometry is applicable to measures of the tendon. Other papers use myotonometry for stiffness measurements using the Myoton, but we hope that this new term tendotonometry can be applied in literature for all tendon stiffness measurements performed with the palpation device.

4. The authors need to expand on the gap in knowledge in the introduction, why is it important to take measurements in different locations on the PT? Why is it important to distinguish between recreational athletes and non-athletes? Also Zhu 2022 did not include females in their analysis, you could highlight that this is also part of the gap in knowledge.

4. We thank the reviewer for mentioning this. We agree that the gap in knowledge could be further elaborated. We have changed the introduction in such a way that we think the knowledge gap is more clearly stated. We added: “In most studies performed on PT stiffness, measurements of stiffness were done at the midpoint of the PT [35, 40-44], or the exact measurement location was not mentioned [45]. To the best of our knowledge, only two other studies investigated variation in stiffness across the PT at other locations [46, 47]. Although both studies also investigated PT stiffness using tendotonometry, only one investigated the reliability of this method, and did so at only three locations (proximal-middle-distal) [46]. Additionally, athletes and females were excluded [46]. While the other study included athletes (gender unknown) and measured at six different locations below each other from proximal to distal, no reliability analyses were performed [47]. Altogether, stiffness and reliability data on a wider area across the PT (also in medial and lateral directions) is still lacking for males and females separately. Since the prevalence of patellar tendinopathy, especially at the proximal region of the tendon, is higher in the athletic population, it is important to know how tendon stiffness varies in healthy athletes and if this can be reliably measured across the tendon. If so, tendon stiffness can be monitored at the exact location of the injury to identify changes compared to the healthy situation, and to understand how injury develops.”, in line 68-80.

5. The hypothesis needs to be clearly stated.

5. We indeed saw that the hypothesis was missing in the manuscript. We therefore added the hypothesis at the end of the introduction: “We hypothesize that i) PT stiffness can be measured reliably at different locations across the tendon, and ii) that PT stiffness varies across the tendon, with an increasing stiffness closer to the insertion.”, line 82-84.

6. Line 80-83. “Measurements took place before any sports activities were performed by the athletes.” For that specific day? If the participants exercised, how long after exercise would the researchers wait to take measurements, were participants instructed not to exercise for x-hours/days prior to testing?

6. Measurements took place with participants not having exercised at least one hour. We have added these details in the manuscript, see line 89-90: “Athletes did not perform any sports activities at least within one hour of measurements.”.

7. Line 93. What medications were a part of your exclusion criteria? Does this include medications like oral contraceptives, which have been shown to alter tendon mechanics?

7. Oral contraceptives were not excluded, since this would exclude a big part of the female athlete population. However, athletes were excluded for example when using medication for a musculoskeletal disorder. We have included the inclusion of oral contraceptive users in the limitation section of the discussion: “A third limitation is that athletes using oral contraceptives were not excluded from participation, since this would have led to the exclusion of a large part of the female athlete population. However, it is known that oral contraceptives can influence tendon properties and collagen synthesis, and therefore this may have influenced our findings [72].”, line 300-303.

8. How did you determine your sample size? Please provide necessary information.

8. We indeed forgot to mention the sample size calculation. We have added this information in the statistical analyses section in the methods, in line 151-154: “A priori type of power analysis and sample size calculation were performed with G*Power (version 3.1.9.7). The sample size of 22 per group (males and females separately) was calculated in order to achieve a power (1-β err prob) of 0.95 with an effect size of 0.25 and α=0.05, with a predefined minimal detectable change (MDC) of 40.3N/m [51].”.

9. Line 92. Given units for BMI (kg/m2). Was the stated BMI range an exclusion criteria? If yes, 25 kg/m2 is a very low cut-off value for BMI even in a healthy population. Please explain.

9. Indeed, the stated BMI range was an exclusion criteria. According to the World Health Organization, a BMI ranging between 18.5 and 24.9kg/m2 is seen as normal or healthy, whereas a BMI below 18.5kg/m2 can be seen as underweight and a BMI above 24.9kg/m2 counts as pre-obesity [1]. Since obesity is related to an increased risk of tendinopathy, we chose this WHO defined healthy BMI range [2].

10. Lines 139-142 should be before “statistical significance was set at p<0.05”, lines 137-138.

10. We have relocated the sentence ‘statistical significance was set at p<0.05’ after the section 160-164: “To investigate the difference in stiffness within the nine different locations, the average PT stiffness of the threefold measurements per location was used in the one-way ANOVA. Tukey HSD test was used as post-hoc test to test for specific group differences. These differences were investigated only for the dominant PT and separately for males and females, since PT stiffness is known to significantly differ between sexes [51]. Statistical significance of all tests was set at p<0.05.”.

11. Lines 145-146, “Males and females were combined together, since literature showed no difference in gender with respect to reliability of the measurement [50].” The authors could provide ICC analysis for both sexes, since the previous publication did not assess different locations on the PT and you’ve already provided some separate analysis of sex. When assessing biological differences, use sex rather than gender.

11.We thank the reviewer for noting this. We have performed additional ICC analyses for the separate sexes and for completeness have added these results to the supporting information as sub-tables of S3. In addition, we have changed the word ‘gender’ into ‘sex’ throughout the manuscript.

12. Table 2. Provide units, identify acronyms, tests used. Please report on the statistical test results. Publications have obligations to provide as much detail as possible so that your results can (potentially) be used for future reviews and field dissemination.

12. We have added the units and explained the abbreviations in the manuscript. Which test used (One-Way ANOVA) is stated in the methods section, but is also added to the table. The statistical test results of all comparisons between the nine different measurement locations are added to the supporting information, performed using a Tukey HSD.

13. Lines 160-163. Same as comment #1, need some edits for readability especially when acronyms are being used. Example: both proximal and distal stiffness were greater than HM, and VM stiffness was greater than medial and lateral.

13. We have tried to rewrite the section accordingly and we hope it is easier to read with the revised version: “In both males and females, PT stiffness was higher at both the proximal and distal sites compared to the HM. PT stiffness at the VM was higher than the medial and lateral regions (Table 2 and Table S2a and S2b in supporting information). The difference in stiffness between the HM and proximal was larger than between HM and distal. This pattern was not observed when comparing VM with medial and lateral regions.”, lines 182-186.

14. Table S2 could and probably should be presented as Table 3 within the results rather than supporting information. Also recheck ICC values, line 168 states all values were above 0.963, in the table it is 0.962. Perhaps a typo?

14. We have added the table to the manuscript, as indeed, it displays one of the main answers to the research question. The ICC results for males and females separately (as asked in #11) are in the supporting information. We would like to thank the reviewer for noticing the typo, which is corrected now.

15. Lines 169 to 170. This is a little misleading, all cases were not <3%; in S3 you state a % of cases >3%. More clarity needed here. I also suggest adding S3 to the main body of the paper as well, or at least combining it with the previous table?

15. We have added table S3 to the manuscript as table 4. In addition, we have clarified the number of cases above 3%. The average of CV were all well below 3%, indeed not the separate cases. We have rewritten the section, see lines 194-195: “With respect to the CV, all averages were well below 3%, with 79% or more separate cases below 3% (Table 4)."

16. Lines 172-176. Once hypothesis is clarified (#5), a statement here would be appropriate.

16. We have added the hypothesis to the first discussion section: “The current study aimed to investigate differences in PT stiffness and reliability of tendotonometry at various locations on the PT in recreational athletes. In line with our hypothesis, this study showed that PT stiffness was higher at proximal and distal ends compared to HM and lower at the medial and lateral sides compared to VM. We also observed that PT stiffness was higher at the proximal site compared to distal. Our findings also support tendotonometry as a reliable method to measure PT stiffness at different locations on the tendon.”, lines 201-206.

17. Lines 188-189, broaden argument about why recreational athletes would have altered stiffness compared to non-athletes.

17. We have added a sentence about the difference in PT stiffness between athletes and non-athletes, and the fact that training can alter PT stiffness. We hope the argument is now written more clearly and thoroughly: “This could be due to the different study population (athletes versus non-athletes [52-54]) and the different measurement locations (at 10% or at the proximal and distal ends of the tendon). Although the direction of change remains unclear, it is known that athletes have an altered PT stiffness compared to non-athletes, and that one single training session can already change tendon stiffness [33]. Loading can lead to adaptations in tendon stiffness by stimulating collagen synthesis and crosslinking between collagen molecules [55-57], making the tendon stronger and, increasing the load capacity.”, lines 218-223.

18. Line 150 “Maximum CV allowed for tendotonometry was preset at 3%”. I assume the authors would retake the exam if the measure was >3%? This adds confusion to lines 224-225 “mean CV below 3%”, the mean CV is always going to be below 3% because of the cut off value. There needs to be more clarity regarding CV and the 3% threshold.

18. We understand the confusion of the reviewer. However, since retaking measurements would definitely influence study outcomes, we have not performed any new measurements. We first performed all the measurements, and afterwards calculated the CV and counted the cases above 3%. The set threshold of 3% was therefore only set for the statistical analysis and not for the study measurements. We have rewritten the section in the statistical methods to enlighten this: “A maximum CV of 3% was defined as good

---

## [Decision Letter · Decision Letter 1]

17 Jul 2025

PONE-D-25-19089R1Differences in stiffness across the patellar tendon: an observational study using tendotonometry.PLOS ONE

Dear Dr. van Dam,

Thank you for submitting your manuscript to PLOS ONE. After careful consideration, we feel that it has merit but does not fully meet PLOS ONE’s publication criteria as it currently stands. Therefore, we invite you to submit a revised version of the manuscript that addresses the points raised during the review process.

**ACADEMIC EDITOR:** The reviewers and I deem the manuscript to be acceptable for publication, however I want to give you the opportunity to review the recent work highlighted by reviewer 1, and make amendments to your work if you choose. You would not have the opportunity to do so if I accept the manuscript at this stage. If you do not wish to incorporate the final comment in to your manuscript, you may resubmit the manuscript without changes.

We look forward to receiving your revised manuscript.

Kind regards,

Charlie M. Waugh

Academic Editor

PLOS ONE

Journal Requirements:

Reviewers' comments:

Reviewer's Responses to Questions

**Comments to the Author**

1. If the authors have adequately addressed your comments raised in a previous round of review and you feel that this manuscript is now acceptable for publication, you may indicate that here to bypass the “Comments to the Author” section, enter your conflict of interest statement in the “Confidential to Editor” section, and submit your "Accept" recommendation.

Reviewer #1: All comments have been addressed

Reviewer #2: All comments have been addressed

2. Is the manuscript technically sound, and do the data support the conclusions?

Reviewer #1: Yes

Reviewer #2: Yes

3. Has the statistical analysis been performed appropriately and rigorously? 

Reviewer #1: Yes

Reviewer #2: Yes

4. Have the authors made all data underlying the findings in their manuscript fully available?

Reviewer #1: Yes

Reviewer #2: Yes

5. Is the manuscript presented in an intelligible fashion and written in standard English?

Reviewer #1: Yes

Reviewer #2: Yes

6. Review Comments to the Author

Reviewer #1: Thank you for the thoughtful responses, good work on the revisions!

Since reviewing this paper, there is a new publication that doesn't find a relationship between tensile stiffness and the MyotonPRO's compressive stiffness (Relationship between compressive stiffness and tensile stiffness in the human Achilles tendon in vivo. Ishigaki, Tomonobu et al. Journal of Bodywork and Movement Therapies, Volume 42, 1073 - 1078). My last revision suggests referencing this paper as it allows the author to be a little more decisive with their language in lines 303-307.

Reviewer #2: (No Response)

7. PLOS authors have the option to publish the peer review history of their article (what does this mean? ). If published, this will include your full peer review and any attached files.

**Do you want your identity to be public for this peer review?** For information about this choice, including consent withdrawal, please see our Privacy Policy .

Reviewer #1: No

Reviewer #2: No

---

## [Author Response · Author response to Decision Letter 2]

18 Jul 2025

Response to editor and referees of manuscript PONE-D-25-19089

Differences in stiffness across the patellar tendon. An observational study using tendotonometry.

Dear editor,

Thank you for offering the opportunity to revise the manuscript entitled “Differences in stiffness across the patellar tendon. An observational study using tendotonometry.” (PONE-D-25-19089). We are thankful to the reviewers for their useful and constructive feedback, which, we believe, has further improved our manuscript.

Below you will find our reply to the comments, which have been incorporated into the revised manuscript. We are submitting the revised manuscript using tracked changes.

Thank you for your time and effort to re-evaluate our revised manuscript.

We look forward to your decision.

Yours sincerely,

Also on behalf of the co-authors,

Lotte van Dam, MSc

Reviewer #1:

1. Since reviewing this paper, there is a new publication that doesn't find a relationship between tensile stiffness and the MyotonPRO's compressive stiffness (Relationship between compressive stiffness and tensile stiffness in the human Achilles tendon in vivo. Ishigaki, Tomonobu et al. Journal of Bodywork and Movement Therapies, Volume 42, 1073 - 1078). My last revision suggests referencing this paper as it allows the author to be a little more decisive with their language in lines 303-307.

1. We thank the reviewer for this feedback to improve the manuscript with more scientific references. We have incorporated the mentioned study in our strength and limitations section, in lines 303-312: “Finally, PT stiffness measurements using the MyotonPRO quantify tendon stiffness perpendicular to the PT, while the actual tendon and its stiffness run longitudinal. We are not certain how compressive, transverse measures of stiffness relate to the ability of tendons to withstand tensile deformation. Resent evidence might suggest that compressive stiffness measured using the MyotonPRO does not reflect the tendon’s tensile stiffness under load, as Ishigaki et al. (2025) found no significant relationship between the two measures in the achilles tendon [73]. However, as this study compared measurements in rest (compressive stiffness) with measurements during exercise (tensile stiffness) and these measurements took place in a different tendon, extrapolation to the PT should be done with caution. Nevertheless, this study is proof that care is necessary when drawing conclusions about longitudinal versus perpendicular tendon stiffness.”

---

## [Editor Report · Decision Letter 2]

22 Jul 2025

Differences in stiffness across the patellar tendon: an observational study using tendotonometry.

PONE-D-25-19089R2

Dear Dr. van Dam,

We’re pleased to inform you that your manuscript has been judged scientifically suitable for publication and will be formally accepted for publication once it meets all outstanding technical requirements.

Kind regards,

Charlie M. Waugh

Academic Editor

PLOS ONE
---

## [Editor Report · Acceptance letter]

PONE-D-25-19089R2

PLOS ONE

Dear Dr. van Dam,

I'm pleased to inform you that your manuscript has been deemed suitable for publication in PLOS ONE. Congratulations! Your manuscript is now being handed over to our production team.

Kind regards,

on behalf of

Dr. Charlie M. Waugh

Academic Editor

PLOS ONE